# MicroRNA Expression Profile in Acute Ischemic Stroke

**DOI:** 10.3390/ijms26020747

**Published:** 2025-01-17

**Authors:** Shraddha Mainali, Gaurav Nepal, Kirill Shumilov, Amy Webb, Paolo Fadda, Darya Mirebrahimi, Mohammad Hamed, Patrick Nana-Sinkam, Bradford B. Worrall, Daniel Woo, Nicholas Johnson

**Affiliations:** 1Department of Neurology, Virginia Commonwealth University, Richmond, VA 23298, USA; 2Department of Internal Medicine, Maharajgunj Medical Campus, Tribhuvan University, Kathmandu 44600, Nepal; drgauravnepal@gmail.com; 3Department of Neurosurgery, Virginia Commonwealth University, Richmond, VA 23298, USA; 4Biomedical Informatics Shared Resources, College of Medicine, The Ohio State University, Columbus, OH 43210, USA; 5Genomics Shared Resource, Comprehensive Cancer Center, The Ohio State University, Columbus, OH 43210, USA; 6College of Medicine, Virginia Commonwealth University, Richmond, VA 23298, USA; mirebrahimid@vcu.edu; 7Department of Neurology, Division of Stroke and Neurocritical Care, The Ohio State University, Columbus, OH 43210, USA; 8Department of Internal Medicine, Division of Pulmonary Disease and Critical Care Medicine, VCU School of Medicine, Richmond, VA 23298, USA; 9Department of Neurology, University of Virginia, Charlottesville, VA 22903, USA; 10Department of Neurology and Rehabilitation Medicine, University of Cincinnati College of Medicine, Cincinnati, OH 45267, USA

**Keywords:** acute ischemic stroke, miRNA, biomarker, micro-RNA, miR, large vessel occlusion, LVO, stroke biomarker

## Abstract

Acute ischemic stroke with large vessel occlusion (LVO) continues to present a considerable challenge to global health, marked by substantial morbidity and mortality rates. Although definitive diagnostic markers exist in the form of neuroimaging, their expense, limited availability, and potential for diagnostic delay can often result in missed opportunities for life-saving interventions. Despite several past attempts, research efforts to date have been fraught with challenges likely due to multiple factors, such as the inclusion of diverse stroke types, variable onset intervals, differing pathobiologies, and a range of infarct sizes, all contributing to inconsistent circulating biomarker levels. In this context, microRNAs (miRNAs) have emerged as a promising biomarker, demonstrating potential as biomarkers across various diseases, including cancer, cardiovascular conditions, and neurological disorders. These circulating miRNAs embody a wide spectrum of pathophysiological processes, encompassing cell death, inflammation, angiogenesis, neuroprotection, brain plasticity, and blood–brain barrier integrity. This pilot study explores the utility of circulating exosome-enriched extracellular vesicle (EV) miRNAs as potential biomarkers for anterior circulation LVO (acLVO) stroke. In our longitudinal prospective cohort study, we collected data from acLVO stroke patients at four critical time intervals post-symptom onset: 0–6 h, 6–12 h, 12–24 h, and 5–7 days. For comparative analysis, healthy individuals were included as control subjects. In this study, extracellular vesicles (EVs) were isolated from the plasma of participants, and the miRNAs within these EVs were profiled utilizing the NanoString nCounter system. Complementing this, a scoping review was conducted to examine the roles of specific miRNAs such as miR-140-5p, miR-210-3p, and miR-7-5p in acute ischemic stroke (AIS). This review involved a targeted PubMed search to assess their influence on crucial pathophysiological pathways in AIS, and their potential applications in diagnosis, treatment, and prognosis. The review also included an assessment of additional miRNAs linked to stroke. Within the first 6 h of symptom onset, three specific miRNAs (miR-7-5p, miR-140-5p, and miR-210-3p) exhibited significant differential expression compared to other time points and healthy controls. These miRNAs have previously been associated with neuroprotection, cellular stress responses, and tissue damage, suggesting their potential as early markers of acute ischemic stroke. This study highlights the potential of circulating miRNAs as blood-based biomarkers for hyperacute acLVO ischemic stroke. However, further validation in a larger, risk-matched cohort is required. Additionally, investigations are needed to assess the prognostic relevance of these miRNAs by linking their expression profiles with radiological and functional outcomes.

## 1. Introduction

Acute ischemic stroke (AIS) is a medical emergency characterized by the sudden blockage of blood flow to the brain, resulting in high morbidity and mortality rates worldwide [1]. Emergent treatment decisions are time-critical, necessitating precise determination of stroke onset time or its practical surrogate, the ‘last known well’ (LKW) time, to expedite the initiation of appropriate interventions [2]. Over 30% of AIS cases involve large vessel occlusion (LVO), particularly in major arteries such as the internal carotid artery (ICA) and the anterior (ACA), middle (MCA), and posterior cerebral arteries, significantly contributing to the burden of stroke due to the large area of ischemic tissue and infarction [3].

According to current clinical guidelines, patients presenting with AIS within 4.5 h from symptom onset are candidates for intravenous (IV) thrombolysis. Initiating IV thrombolysis within this timeframe increases the likelihood of improved functional outcomes across all age groups, with the magnitude of benefit being highly time-dependent [4,5]. For patients with anterior circulation large vessel occlusion (acLVO), endovascular thrombectomy (ET) is typically indicated within 6 h of onset without the need for advanced perfusion imaging. Beyond this window, up to 24 h, ET is considered based on a thorough risk/benefit assessment [4]. Contemporary stent retriever devices can achieve successful recanalization in over 87% of patients, significantly enhancing outcomes with a number needed to treat (NNT) of 8 for an excellent clinical outcome and an NNT of 3 for a favorable functional outcome without markedly increasing mortality or hemorrhagic complications [6]. It is estimated that increasing the rate of near-complete to complete reperfusion by just 10% could result in an additional 3656 quality-adjusted life years (QALYs) and save USD 21.0 million and USD 36.8 million for the US healthcare system and society, respectively [7]. Without timely intervention, the progression of stroke leads to rapid loss of neural tissue. In LVO patients, an estimated 120 million neurons, 830 billion synapses, and 714 km (447 miles) of myelinated fibers are lost every hour, leading to progressive neurological decline [8]. As the stroke advances, the risk of intracranial hemorrhage (ICH) begins to outweigh the benefits of recanalization therapy, typically beyond 24 h [9,10]. Therefore, the determination of stroke onset time is paramount in delivering safe and effective treatment for stroke patients.

Nevertheless, a substantial proportion of AIS patients, approximately one in four, present with unclear stroke onset time or LKW, making them ineligible for potentially life-saving acute stroke intervention [11]. The lack of reliable methods to estimate the time of stroke onset and accurately gauge the extent of tissue injury poses a significant challenge in managing AIS effectively, especially in cases where the precise timing of symptom onset remains uncertain.

The recent literature has highlighted the significance of molecular intercellular messaging and signaling in determining the state of tissue injury in various diseases, including stroke [12]. MiRNAs have emerged as a class of non-coding RNA molecules that play a pivotal role in intercellular communication by regulating the expression of target mRNAs [13]. In the context of AIS, miRNAs have shown promise as potential biomarkers for diagnostic and prognostic applications [14,15,16,17]. Studies have reported altered miRNA expression profiles in blood and brain tissues of AIS patients, suggesting their potential as biomarkers for stroke detection, subtyping, and prognosis [18,19]. However, existing studies on stroke have reported variable findings due to heterogeneity in study populations, including variations in lesion distribution, timing from symptom onset to sample collection, and differences in the biological sources of miRNAs (e.g., cells, tissue, serum, plasma). Additionally, inconsistencies in sample processing methods further contribute to this variability. To harness the full potential of miRNAs as biomarkers, it is crucial to study their expression systematically in specific stroke types, enabling the identification of meaningful and reliable expression signals. The primary objective of this pilot study was to investigate circulating miRNA expression profiles specifically during the hyperacute phase (within 6 h of symptom onset), focusing on acLVO stroke, which is associated with high morbidity and mortality but has the potential for significant improvement with timely acute stroke interventions. Furthermore, to date, no study has comprehensively evaluated the temporal miRNA expression profiles in hyperacute AIS patients with acLVO stroke over 7 days following symptom onset. This pilot study provides critical insights into the dynamic shifts in circulating miRNA expression patterns related to hyperacute acLVO stroke, observed longitudinally over a week. Understanding the dynamic changes in miRNA expression during the hyperacute phase of acLVO would be highly beneficial for unraveling the early molecular signals underpinning this devastating condition. This knowledge opens avenues for potential blood-based biomarkers that could transform early diagnosis and monitor treatment efficacy. Moreover, it sheds light on the molecular dynamics of stroke progression and tissue damage, offering opportunities for improved clinical decision-making, prognostication, and the discovery of new therapeutic targets [20]. In this article, the findings of the pilot study are contextualized through a scoping review of the existing literature to identify knowledge gaps, assess the relevance of our results, and inform future research directions.

## 2. Results

We aimed to identify differentially expressed miRNAs that could provide key insights into the pathophysiology of acLVO stroke. Analysis was performed on a total of 24 samples from six patients with confirmed acLVO stroke, with 12 samples from patients presenting with right ICA/MCA involvement and the remaining 12 samples with left ICA/MCA involvement. Individual patient characteristics are presented in Table 1. All patients received IV thrombolytic therapy (tPA) within the appropriate therapeutic window. Furthermore, some patients underwent endovascular thrombectomy as an additional intervention to restore blood flow, resulting in the collection of plasma samples during (one patient) and after the thrombectomy procedure (two patients). Control included five healthy volunteers (three males, two females) between the ages of 18 and 60 years.

The ANOVA analysis of the raw data revealed statistically significant differential expression of 11 miRNAs (miR 210-3p, miR 7-5p, miR 122-5p, miR 140-5p, miR 378i, miR-320e, miR 448, miR 1258, miR 26a-5p, miR 28-5p, miR 510-3p) across various time points and healthy volunteer comparisons. Recognizing the clinical importance of the hyperacute period within 6 h of stroke onset to understand the early molecular response to ischemia, our study concentrated on identifying specific miRNAs that exhibit unique expression profiles during this initial phase in contrast to subsequent time points and healthy volunteers. Notably, three miRNAs, miR 140-5p, miR 7-5p, and miR 210 3p (Figure 1), exhibited significant differential expressions within the first 6 h compared to both the healthy volunteers and other time points.

Particularly intriguing was the observation that miRNA 140-5p displayed noticeable elevation within 6 h, gradually normalizing between 12 and 24 h and ultimately approaching the volunteer level after 7 days (Figure 2). Similarly, miRNA 7-5p exhibited clear overexpression within the first 6 h, followed by a gradual downtrend towards the volunteer level by 24 h, maintaining stable expression within a similar range at 7 days. In contrast, miRNA 210-3p demonstrated under-expression at 6 h, gradually increasing towards the volunteer level over the next 12–24 h and maintaining that level through 7 days.

In our Ingenuity Pathway Analysis (IPA) by QIAGEN (Hilden, Germany), comparing enriched downstream pathways across various time points in stroke, we observed multiple pathways activated over time, following stroke, based on messenger RNA targets regulated by the miRNAs identified in each time comparison (Figure 3). Within the first 6 h post-stroke, distinct signaling pathways emerge that may contribute to the early pathophysiology of ischemic stroke, including inflammation, cell death, and thrombosis. TNFR1 signaling and TWEAK signaling are prominent during this initial phase, both of which are associated with inflammation and apoptosis [21]. TNFR1 signaling can initiate the release of pro-inflammatory cytokines and promote cell death, while TWEAK signaling, as part of the TNF superfamily, also contributes to inflammatory responses and has been linked to blood–brain barrier disruption and edema formation; critical processes in the early response to ischemic injury. Additionally, GP6 signaling, a pathway involved in platelet activation [22], is observed, highlighting the role of platelet aggregation and thrombus formation, which are central to the development and progression of ischemic events. Compared to later time points, where pathways such as IL-17 signaling become prominent, the first 6 h are marked by immediate inflammatory and thrombotic responses, setting the stage for subsequent sustained inflammation and immune activation over the following days. The appearance of the Hepatic Fibrosis/Hepatic Stellate Cell Activation pathway in this timeframe may indicate the activation of shared signaling pathways involved in inflammation and tissue remodeling.

Our IPA analysis highlights a potential linkage between the early activation of inflammatory and thrombotic pathways, such as TNFR1, TWEAK, and GP6 signaling, and the three candidate circulating miRNAs, miR-140-5p, miR-210-3p, and miR-7-5p, identified within the first 6 h of large vessel occlusion (LVO) stroke. Preclinical studies in stroke models suggest that miR-140-5p and miR-7-5p may modulate early inflammatory and vascular responses, aligning with the roles of TNFR1 and TWEAK in regulating inflammation, apoptosis, and edema formation [23,24,25,26]. Similarly, miR-210 has shown potential for influencing long-term inflammation and recovery processes, which is consistent with sustained inflammatory pathways like IL-17 and hepatic fibrosis signaling observed in IPA. These findings support the role of these miRNAs in reflecting and potentially modulating key inflammatory and immune responses in acute stroke.

### Review of Current Knowledge on miR-140-5p, miR-210-3p, and miR-7-5p in Acute Ischemic Stroke

**MiR 140-5p:** A concise review of the existing literature regarding the role of these three miRNAs in AIS is presented in Table 2. In animal models simulating ischemic stroke, a significant decrease in miR-140-5p expression was observed within the ischemic core [23,24,25]. Conversely, when examining human serum samples, all studies consistently reported an elevation in serum miR-140-5p levels after cerebral ischemia [27,28,29]. This alignment with our findings suggests the potential utility of miR-140-5p as a diagnostic marker, a notion supported by these consistent outcomes. The observed elevation of miR-140-5p in circulation and its concurrent reduction within the ischemic brain tissue during the initial 6-h window may reflect a response to ischemic insult. Ischemia/reperfusion injury might trigger the translocation of miR-140-5p from the affected neurons to the circulatory system. This process could be related to the mobilization of inflammatory mediators and growth factors, crucial for the brain’s intrinsic response to ischemic damage.

Moreover, studies have also explored the therapeutic potential of miR-140-5p. Wang et al. demonstrated that administration of encapsulated miR-140-5p could alleviate neuronal damage in subarachnoid hemorrhage [23]. Liang et al.’s work showcased that overexpressing miR-140-5p using adeno-associated viruses reduced inflammatory and vascular growth factors in the ischemic mouse hippocampus, inhibiting neurogenesis and capillary density [27]. Similarly, Sun et al. revealed that miR-140-5p hinders angiogenesis after cerebral ischemia, potentially contributing to the mitigation of hemorrhagic transformation and edema [24]. Additionally, Song et al. provided evidence that miR-140-5p overexpression inhibited neuron apoptosis and decelerated stroke progression [25]. While these animal model and in vitro studies show promise for the therapeutic role of miR-140-5p, it is important to note that the limited number of studies and inconsistencies in miR-140-5p delivery methods preclude any definitive conclusions.

**MiR 7-5p**: As illustrated in Table 2, our literature search found two human-based studies regarding the role of miR-7-5P as a biomarker in stroke. In contrast to our study, Ni et al. observed a reduction in miR-7-5p levels following stroke. However, in contrast to our study, they did not detail the precise timing of sample collection, referring instead to a broader 48-h window [30]. Meanwhile, Chen et al. demonstrated that in humans with intracerebral hemorrhage (ICH), the serum levels of miR-7-5p were significantly higher on day one compared to day 7, demonstrating time-dependent evolution in ICH [26]. Most studies in animal models of cerebral ischemia and intracerebral hemorrhage have indicated a decrease in miR-7-5P levels in brain tissue samples [26,30,31,32]. Similarly, in a model of carotid artery injury, miR-7-5p was found to be downregulated when examining carotid endarterectomy samples [31,33]. Similar to 140-5p, the decreases in tissue miR-7-5p levels might be attributed to the release of miR-7-5P from injured tissue into the serum. However, Zhao et al. observed a contrasting trend, with miR-7-5P significantly upregulated in ischemic brain tissue in a time-dependent manner [34]. Dharap et al. noted no initial change in miR-7-5P levels, followed by a decline after 12 h in a rat model of focal ischemia [35]. Given these conflicting results in varied models with varied tissue types and sampling time points, the utility of circulating miR-7-5P as a diagnostic biomarker remains uncertain and needs further evaluation.

**Table 2 ijms-26-00747-t002:** Essential methodological aspects and insights from stroke literature investigating the function of three principal miRNAs identified in our study.

miR	Author	Model Studied/Disease	Sample Tissue	Sample	Collection Time	Major Findings	Function/Pathway
miR 140-5P	Wang 2022 [23]	Rats/in vitro Endovascular perforation models of SAH	Brain tissue	93	3rd day	Exoencapsulated miR-140-5p can relieve neuronal injury following SAH.	Modulation of the IGFBP5-mediated PI3K/AKT signaling pathway.
	Liang 2019 [27]	Human Post-stroke depression	Plasma	252	Within 24 h	MiR-140-5p (*p* = 0.0016, log2 (fold change) = 3.5) had significantly higher expression in the late-onset PSD group than in controls. The miR-140-5p expression on admission was significantly positively correlated with Hamilton Depression Rating Scale assessed at 3 months after stroke. The predictive value of miR-140-5p for late-onset PSD is 83.3% sensitivity and 72.6% specificity (AUC = 0.8127).	Regulate IL1rap, IL1rapl1, VEGF, and MEGF10.
	Sørensen 2014 [28]	Human Acute Ischemic Stroke	CSF and blood	10 cases and control each	NA	Blood: miR-140-5p (*p* = 0.02) was upregulated in stroke patients compared to controls. CSF: Not detected.	NA
	Song 2021 [25]	Rats/in vitro Acute Ischemic Stroke	Brain tissue	60 cases and 15 controls	NA	miR-140-5p exhibited decreased expression, while TLR4 displayed increased expression. MiR-140-5p directly targeted and reduced TLR4 expression. MiR-140-5p over-expression inhibited neuron apoptosis and slowed stroke progression. TLR4 over-expression promoted neuron apoptosis and stroke progression. MiR-140-5p reduced NF-κB protein levels, while TLR4 overexpression increased them.	Regulation of the TLR4/NF-κB axis.
	Toor 2022 [29]	Humans/in vitro Acute Ischemic Stroke	Serum	190	Within 24 h	miR-140-5p was observed to be upregulated in stroke patients with diabetes.	Regulate genes involved in inflammation and oxidative stress.
	Sun 2016 [24]	Rats/in vitro Acute Ischemic Stroke	Brain tissue	24 cases and 8 control	12, 24, 48 h	The expression of miR-140-5p exhibited a significant reduction at 12, 24, and 48 h post-MCAO compared to the control. Conversely, the protein expression levels of VEGFA showed a significant increase at 12, 24, and 48 h following MCAO compared to the control.	Suppresses angiogenesis by targeting VEGFA.
miR 210-3P	Pfeiffer 2021 [36]	Rats/in vitro Acute ischemic stroke	Brain tissue	75	24 h after ischemia	In response to transient focal ischemia with reperfusion, miR-210-3p is upregulated in the cortex. When a miR-210-3p mimic is administered in vivo, it changes the expression of key signaling molecules like PTEN, PDK1, p70S6K, and RPS6. This manipulation also results in a decrease in p70S6K activity following an ischemic stroke. miR-210-3p influences p70S6K activity in response to NMDA-mediated excitotoxicity, and this effect can be reversed by inhibiting miR-210-3p. Pre-treatment with 5 pmol miR-210-3p mimic resulted in a significant decrease in hemispheric swelling and infarct volume.	AMPK regulates miR-210-3p to control p70S6K activity.
	Rahmati 2021 [37]	Humans Acute ischemic stroke	Serum	52 cases	Admission, 24 and 48 h after admission, upon discharge, and 3 months later	Serum miR-210 levels in cases were initially lower upon admission compared to normal controls but increased progressively over three months. A diagnostic cutoff point was set at a fold change of 0.26 with an AUC of 0.61, 59.62% sensitivity, and 65.38% specificity. Higher miR-210 expression at the three-month follow-up was linked to improved survival in IS patients.	NA
	Eken 2016 [38]	Humans and rat model/in vitro Carotid atherosclerosis	CEA tissue and blood samples	Symptomatic humans: 19 Rats: 48	At the time of surgery	MiR-210 is downregulated in symptomatic carotid stenosis patients’ plasma and fibrous cap tissue. It is repressed in experimental artery remodeling and influences plaque stability in atherosclerosis.MiR-210 mimics prevent plaque rupture in vivo and protect smooth muscle cell apoptosis by targeting APC in vitro.	Inhibits APC.
	Ujigo 2014 [39]	Rats/in vitro Spinal cord injury	Spinal cord tissue	30	2, 3, 5, 7, and 14 days after SCI	Hsa-miR-210 upregulated miR-210 expression, leading to enhanced neovascularization, astrogliosis, axon growth, and myelination in the injured spinal cord. In the miR-210 group, there were significantly fewer apoptotic cells at the lesion site, and caspase-3 and cleaved caspase-3 levels were markedly reduced compared to the control group. miR-210 administration promoted functional recovery after spinal cord injury	Inhibits Ptpib and Efna3.
	Zeng 2016 [40]	Human/ Rats/in vitro AIS	Serum for humans Brain tissue for rats	Humans: 5 cases, 5 controls. Rats: 124 total, divided into sham, transient MCAO, tMCAO + LV-GFP, and tMCAO + LV-miR-210 groups.	Humans: Within 48 h and the 10th day of AIS Rats: 7, 14, and 28 days after tMCAO	MiR-210 was downregulated in stroke patients vs. healthy controls. MiR-210 gene transfer improved outcomes in tMCAO mice.	BDNF regulation.
	Zeng 2011 [35]	Humans/Rats AIS	Human: Serum Rats: Brain tissue and serum	Stroke patients (n = 112) and healthy controls (n = 60) 9 rats	Human blood: 3, 7, and 14 days post-stroke. Rat blood and brain tissue: 1, 7, and 14 days post-MCAO.	In stroke patients, blood miRNA-210 levels were significantly lower, particularly at 7 and 14 days post-stroke onset compared to healthy controls. MiR-210 rose one day after MCAO in rats, declining gradually at 7 and 14 days, and a significant positive correlation existed between blood and brain miR-210 levels. A diagnostic cutoff point of 0.505 for miR-210 yielded an 88.3% sensitivity. Stroke patients with favorable outcomes exhibited higher miR-210 levels than those with poor outcomes.	NA
	Ma 2021 [41]	Rats/in vitro Neonatal hypoxic-ischemic brain injury	Brain tissue	NA	48 h after HI	In neonatal mouse pups, hypoxic-ischemic (HI) conditions increased miR-210, which suppressed TET2 expression and led to enhanced p65 acetylation and binding at the IL-1β promoter in the brain. TET2’s interacted with HDAC3 regulated NF-κB p65’s DNA binding at the IL-1β gene promoter. TET2 knockdown elevated p65 acetylation, increased pro-inflammatory cytokine and chemokine expression after HI, and worsened neonatal HI brain injury. It also counteracted the anti-inflammatory effect of miR-210 inhibition in neonatal HI brain injury and BV2 microglia cell line experiments in vitro.	TET2 downregulation.
	Li 2023 [42]	Rats/in vitro AIS/MCAO model	Brain tissue	204	NA	miR210 injection reduced TET2 in the brain, but miR210 inhibition or KO preserved TET2 irrespective of brain injury. TET2 reduction reversed miR210 inhibition’s protective effects on stroke-induced brain damage and neurobehavioral deficits. Lowering TET2 weakened miR210’s anti-inflammatory impact on microglial activation and IL-6 release after stroke. Boosting TET2 in microglia counteracted miR210-induced cytokine increase.	TET2 downregulation.
	Yerrapragada 2022 [43]	In vitro Hypoxia and reoxygenation (H/R) models were applied to neurons to mimic AIS	In vitro	NA	NA	Endothelial progenitor cell (EPC)-borne miR-210 can be time-dependently transferred to neurons, exerting a protective impact against H/R-induced neuron apoptosis, oxidative stress, and reduced viability.	BDNF/TrkB and Nox2/Nox4 pathways regulation.
	Zhang 2019 [44]	Rat/in vitro AIS/MCAO model	Brain tissue	NA	24 h after MCAO	RGD-exo:miR-210 targets the ischemic brain lesion upon intravenous delivery, elevating miR-210 levels at the site. Administered every other day for 14 days, it notably boosts the expression of integrin β3, vascular endothelial growth factor (VEGF), and CD34, leading to an improved animal survival rate.	RGD-exo:miR-210 promotes VEGF expression and angiogenesis.
	Huang 2018 [45]	Rats/in vitro AIS/MCAO model	Brain tissue	96-miR-210-LNA treatment (n = 44), negative control (n = 41), other experiments (n = 11)	24 h after MCAO	MiR-210-LNA pre-treatment reduced brain infarct volume edema and improved post-stroke behavior in MCAO mice. It also suppressed pro-inflammatory cytokines, chemokines, immune cell infiltration, and microglial activation induced by MCAO. Posttreatment with MiR-210-LNA was also protective in MCAO mice.	MiR-210-LNA mitigates inflammatory response after cerebral ischemia.
	Tian 2021 [46]	Humans AIS	Serum	76 cases 64 controls	At admission	miR-210 levels were significantly lower in the mortality group compared to the survival group.MiR-210 had high diagnostic accuracy for acute cerebral infarction (AUC = 0.836) and was associated with lower 1-year survival in the low-expression group. It also showed good predictive capability for mortality (AUC = 0.786).	NA
	Lu 2019 [47]	In vitro Endothelial progenitor cells (EPCs) under hypoxic condition	In vitro	NA	NA	In OGD-treated EPCs, miR-210-3p expression was higher than in normal EPCs. Increased miR-210-3p enhanced proliferation, migration, and tube formation under OGD conditions, while decreased miR-210-3p hindered these capabilities in OGD-treated EPCs. Elevated miR-210-3p suppressed repulsive guidance molecule A (RGMA) protein expression in OGD-treated EPCs, while reduced miR-210-3p led to increased RGMA expression.	Inhibits RGMA, a negative regulator of angiogenesis.
miR 7-5P	Chen 2020 [26]	Humans and rats/in vitro Intracerebral hemorrhage (ICH)	Serum in humans Brain tissue in rats In vitro	60 rats	1, 7, and 14 days after ICH	Humans: The miR-7-5p level decreased significantly on day 7 after ICH compared to day 1 but showed partial recovery by day 14. MiR-7-5p expression significantly decreased on days 1, 3, and 7 after ICH, with the most pronounced decrease on day 3. Partial recovery occurred after butylphthalide intervention. The brain water content decreased in the butylphthalide group.	PI3K/AKT pathway regulation.
	Xu 2019 [31]	Rats Ischemic stroke	Brain tissue/In vitro		NA	Curcumin prevented the decrease of miR-7-5p expression and the increase of RelA p65 expression caused by cerebral ischemia–reperfusion injury (CIR) in vivo and oxygen-glucose deprivation/reoxygenation (ODG/R) in vitro. MiR-7-5p was found to target RelA p65. MiR-7-5p antagonists reversed curcumin’s impact on RelA p65 expression in ischemic brain tissue and cells.	Curcumin regulates miR-7/RELA p65 axis.
	Kim 2018 [32]	Rats Ischemic stroke	Brain tissue/in vitro	NA	Day 7 and 31	Ischemia reperfusion-induced a 2.1- to 3-fold decrease in miR-7 expression in the ipsilateral cortex of both young and aged rats of both sexes compared with the sham-operated controls. Preischemic intracerebral administration of miR-7 mimics improved motor function recovery and decreased lesion volume in young male rats. Postischemic intracerebral administration of miR-7 mimics decreased ischemic brain damage irrespective of sex and age in rats. Postischemic intravenous administration of miR-7 mimic decreased the stroke-induced cognitive deficit and accelerated motor recovery but failed to do so in alpha syn knockout mice.	α-Synuclein regulation.
	Dharap 2009 [35]	Rats Ischemic stroke	Brain tissue	30 cases and 6 controls	3, 6, 12, 24, and 72 hr	Transient focal ischemia in a rat model induced no change in miR 7-5P in the first 12 h, followed by a sustained decrease.	NA
	Zhao 2020 [34]	Rats Ischemic stroke	Brain tissue/in vitro	36	24 h after reperfusion	MiR-7-5p exhibited high expression in a rat model with cerebral I/R injury and OGD/R-induced SH-SY5Y cells increasing progressively with time. Silence of miR-7-5p impaired ischemia-reperfusion injury Caused Cerebral Injury. Attenuation of miR-7-5p Hindered I/R Caused Cerebral Inflammation. Depletion of miR-7-5p impeded neuronal cell apoptosis miR-7-5p regulated neuronal cell apoptosis by Targeting sirt1 MiR-7-5p knockdown blocked the NF-kB pathway	sirt1 regulation.
	Ni 2015 [30]	Human Ischemic stroke	Plasma and brain tissue/in vitro	8 human cases and controls	Brain tissue: 24 h and 96 h after occlusion Plasma: within 48 h of stroke	miR-7c-5p was significantly decreased in the plasma of ischemic stroke patients and experimental animals. There was a significant decrease of let-7c-5p in the ipsilateral cortex and striatum of mice subjected to middle cerebral artery occlusion (MCAO) at 24 h reperfusion. Overexpression of let-7c-5p via ICV injection decreased the infarction volume and attenuated the neurological deficits. miR-7c-5p directly targeted the 30-untranslated region of the caspase 3 mRNA to reduce caspase 3 levels, which may underline the miRNA-modulated microglial activity.	Inhibition of microglia activation.
	Yuan 2023 [33]	Rats Carotid injury	Carotid artery sample/in vitro	NA	After 12 weeks of treatment	MiR-7-5p was downregulated and NF-κB p65 was upregulated in injured carotid arteries in the rat model. MiR-7-5p relieves intimal hyperplasia in carotid artery injury rat model. MiR-7-5p attenuates the proliferation and migration of VSMCs. MiR-7-5p represses NF-κB p65 in VSMCs and inhibits the proliferation and migration of VSMCs.	NF-kB signaling.

Several investigations have focused on the therapeutic implications of miR-7-5p. Chen et al. found that miR-7-5p levels were raised by butylphthalide via intracerebroventricular administration, which contributed to the alleviation of brain edema [26]. Xu et al. reported that curcumin regulates miR-7-5p, conferring neuroprotection and ameliorating cognitive deficits in ischemic reperfusion injury [31]. Kim et al. observed that preischemic administration of miR-7 mimics enhanced motor function and diminished lesion volume in young male rats, while post-ischemic treatment was effective in reducing brain damage across all rats, improving cognitive outcomes and expediting motor recovery [32]. Additionally, Ni et al. demonstrated that elevating let-7c-5p levels via intra-cerebrovascular injection reduced infarct size and lessened neurological impairments [30]. Conversely, Zhao et al., studying a rat model of ischemia reperfusion, identified that an increase in miR-7-5p was associated with heightened inflammation, apoptosis, and the exacerbation of ischemic damage [34]. Overall, the current body of research on miR 7-5p also reveals variations in the miRNA expression profile possibly linked to varied types of biological specimens, disease severity, sampling timepoint, and miRNA profiling techniques.

**MiR 210**: MiR-210 has also received considerable attention in stroke research, as detailed in Table 2, with investigations encompassing in vitro analyses, animal models, and clinical studies to evaluate its diagnostic, prognostic, and therapeutic potential. Across these studies, a recurrent finding is the elevation of miR-210 expression within brain tissue following cerebral ischemia, including ischemic stroke and hypoxic-ischemic encephalopathy [36,40,41,42,44,45,48]. In contrast, circulating levels of miR-210 in ischemic stroke patients appear to be suppressed when compared to those of healthy controls [38,40,41,42,44,45,46,48]. This same trend is observed in patients with symptomatic carotid stenosis, where miR-210 is downregulated in carotid fibrous cap tissue [45]. This finding underscores the potential of miR-210 as a reliable biomarker for cerebral ischemia. Supporting its diagnostic role, Rahmati et al. established a threshold for miR-210 with a fold change of 0.26, correlating with a modest diagnostic performance characterized by an area under the receiver operating characteristic curve (AUC) of 0.61 and exhibiting 59.62% sensitivity and 65.38% specificity [37]. Zeng et al. identified a higher sensitivity at a diagnostic cutoff point of 0.505 for miR-210, achieving 88.3% sensitivity [48]. Complementing these studies, Tian et al. confirmed the high diagnostic accuracy of miR-210 for acute cerebral infarction, presenting an AUC of 0.836 [46]. These findings collectively point toward the potential of miR-210 as an informative biomarker for the identification of acute ischemic events.

The role of miR-210 in prognostication for ischemic stroke patients has been substantiated by multiple studies. For instance, Rahmati et al. found a positive correlation between elevated miR-210 levels at three months post-stroke and enhanced survival rates [37]. Zeng et al. reported that patients with favorable recovery showed higher miR-210 expression than those with adverse outcomes [48]. On the contrary, Tian et al. reported that patients with lower miR-210 expression levels had increased one-year mortality, with miR-210 levels emerging as a robust predictor of mortality (AUC = 0.786) [46]. While the current body of research presents variability, likely attributable to insufficient control of confounding variables across different studies, the aggregated evidence nonetheless points to miR-210 as a potentially valuable marker for predicting neurological outcomes in acute ischemic stroke scenarios.

In terms of therapeutic implications, miR-210 has shown potential in both in vitro and animal models as summarized in Table 2. Research by Eken et al. demonstrated the prophylactic effect of miR-210 mimics on carotid plaque stability, suggesting a preventative role against ischemic stroke [38]. Pfeiffer et al.’s subgroup analysis revealed that pretreatment with a miR-210-3p mimic substantially mitigated hemispheric swelling and infarct size [36]. Similarly, Huang et al. validated the protective effects of miR-210, noting that both pre- and post-treatment with a miR-210 locked nucleic acid (LNA) conjugate led to reduced cerebral infarct and edema, alongside behavioral improvements in mice models of middle cerebral artery occlusion (MCAO) [45]. Additionally, Zeng et al. illustrated the efficacy of miR-210 gene transfer in enhancing recovery in transient MCAO models [40]. Additionally, research by Li et al. and Zhang et al. has highlighted miR-210′s role in attenuating inflammation and reducing ischemic damage in both in vitro settings and cerebral ischemia models [42,44]. Ma et al. demonstrated the neuroprotective effects of exogenous miR-210 mimics in a model of neonatal hypoxic-ischemic brain injury [41], while Lu et al. documented enhanced function of endothelial progenitor cells under hypoxic conditions when treated with miR-210 [47]. Yerrapragada et al. further corroborated the neuroprotective role of miR-210 in a hypoxia and reoxygenation model, indicating its therapeutic potential in mitigating hypoxic-ischemic neuronal damage [43]. Extending beyond cerebral models, Ujigo et al. found that intracranial administration of miR-210 contributed to functional recovery in cases of traumatic spinal cord injury [39]. These studies underscore the promising therapeutic avenues miR-210 may offer for ischemic stroke intervention. Figure 4 presents a summary of the biological pathways associated with miR-140-5p, miR-210-3p, and miR-7-5p.

## 3. Discussion

Our study revealed that, upon comparing each time point against the remaining three time points in acLVO patients and a single time point in healthy volunteers, a total of 11 miRNAs exhibited significantly altered expression across these comparative analyses. Notably, within the first 6 h of acLVO stroke onset, three miRNAs (140-5p, 7-5p, and 210-3p) exhibited significant differential expression compared to healthy volunteers and other time points. MiRNA 140-5p showed a relative increase within the first 6 h, gradually normalizing between 12 and 24 h and reaching volunteer levels within seven days. Similarly, miRNA 7-5p displayed significant overexpression within the first 6 h, followed by a gradual decline towards volunteer levels by 24 h, maintaining stable expression within a similar range around seven days. In contrast, miRNA 210-3p demonstrated relative under-expression at 6 h, gradually increasing towards the volunteer level over the next 12–24 h and maintaining that level through 7 days. These findings highlight the dynamic and time-sensitive nature of miRNA expression in the hyperacute and early post-stroke phases, offering potential biomarkers for monitoring stroke progression and recovery.

Our study and the existing literature have highlighted the significant roles played by miRNAs (140-5p, 7-5p, and 210-3p) in stroke pathophysiology and therapy. Of note, these miRNAs employ diverse mechanisms to exert their effects. We have provided an overview of the various pathways they operate within stroke and related disorders in Table 2. Many of these pathways are closely linked to inflammation, oxidative stress, and edema formation noted in our IPA analysis. Utilizing molecular drug discovery to target these pathways or the miRNAs themselves holds promise as an effective strategy for stroke prevention and treatment. In our study, we have also identified other miRNAs, such as miR 210-3p, miR 122-5p, miR 378i, miR-320e, miR 448, miR 1258, miR 26a-5p, miR 28-5p, and miR 510-3p, in association with ischemic stroke. The functions and potential pathways of these miRNAs and other relevant miRNAs are summarized in Appendix A These miRNAs are subjects of ongoing research in a reverse translational model, aiming to elucidate their roles and mechanisms further.

Our study has several strengths. In this pilot project, we endeavored to meticulously assemble a homogenous cohort of patients, each presenting with anterior circulation acLVO, to maintain uniformity in the stroke phenotype for our analyses. Recognizing the potential for variability introduced by timing, we strictly limited the collection of blood samples to within a 6-h window following the onset of symptoms, which we hoped would reduce confounding factors related to timing ambiguities. We adopted a longitudinal design for the study, which permitted us to cautiously interpret the evolution of miRNA profiles over time, treating each time point as an intrinsic control against the baseline hyperacute samples. This careful approach, while preliminary, was expected to offer valuable insights into the dynamic changes of miRNAs in this context.

While the study presents intriguing outcomes, it is important to recognize its limitations. A key limitation is the modest cohort size, comprising a total of 29 samples, which may limit the generalizability of the findings. The LVO group included a total of four female samples only, which hindered the assessment of potential sex-related differences in miRNA profiles. Although stroke typically occurs in older individuals, the majority of our study’s participants were middle-aged, with one patient being a minor. This distribution may not accurately reflect the age-related risk of stroke in the general population. Additionally, the use of healthy controls who were not matched for stroke risk factors could introduce confounders into the miRNA expression profiles. As Toor et al. indicated, miR-140-5p levels were found to be elevated in stroke patients with diabetes relative to non-diabetic patients [29], suggesting that miRNA expression may differ with underlying risk factors. Moreover, our research was confined to the study of EV encapsulated miRNA, and the potential role of non-vesicular, free-circulating miRNAs was not investigated, which constitutes an area for further research. Furthermore, silencing miRNA could serve as a potential therapeutic approach, as previously demonstrated in a mouse model of lymphoma [49], to prevent additional damage following a stroke.

Our literature review disclosed considerable heterogeneity within the corpus of research investigating the role of miRNAs in ischemic stroke. This variation is likely due to a lack of standardization across several critical aspects of study design and methodology. These aspects include the criteria for control group selection, the source of the miRNAs (serum, plasma, CSF or brain tissue), the protocols used for miRNA isolation, the timing of sample collection (ranging from hyperacute to delayed phases), the selection of reference standards (internal and external controls), the choice of detection and quantification techniques (such as Nanostring, Next-Generation Sequencing, or RT-qPCR), and the breadth of the infarct sizes. Additionally, the biological origin of the miRNAs, whether cellular, vesicular, or cell-free, also contributes to the variability of the results, further complicating the interpretation and comparison of findings across studies.

To enhance the reliability of biomarker studies, future investigations should aim for rigorously matched control groups that align with stroke patients’ symptoms and risk factors, utilizing consistent and validated methodologies within a well-defined stroke cohort. Adopting a multi-center design would improve the robustness and applicability of miRNA biomarkers for diagnostic purposes. Additionally, it is crucial to assess the prognostic value of these miRNAs by examining their associations with both radiological findings and clinical outcomes. Implementing miRNA profiles in the elucidation of disease pathways could inform treatment strategies and support timely consultations with patients and their families. Moreover, in-depth mechanistic research through reverse translational models is needed to decipher the roles of specific miRNAs in the pathogenesis of acLVO strokes, potentially uncovering novel therapeutic avenues. These efforts will deepen our understanding of miRNA-mediated regulation in stroke and could lead to significant advances in patient care.

## 4. Materials and Methods

### 4.1. Study Design

This longitudinal, prospective cohort study, conducted at the Joint Commission-certified Comprehensive Stroke Center of Ohio State University, a tertiary referral medical center, explores the viability of circulating extracellular vesicles (EV)-encapsulated miRNAs as blood-based biomarkers for acLVO ischemic stroke. The study protocol received approval from the Ohio State University Institutional Review Board, and informed consent was secured from all participants or their legally authorized representatives. All methods were performed in accordance with the relevant guidelines and regulations.

Our scoping review was designed to elucidate the roles of certain miRNAs, specifically miR-140-5p, miR-210-3p, and miR-7-5p, within the acute ischemic stroke (AIS) framework. To achieve this, we conducted a comprehensive PubMed search using terms like ‘acute ischemic stroke’, ‘stroke’, ‘miR’, ‘miRNA’, ‘micro RNA’, and the specific miRNAs of interest (miR-140-5p, miR-210-3p, and miR-7-5p). We employed the Boolean operators “AND” and “OR” to refine the search. Our primary objective was to explore the current research on how these miRNAs influence key pathways in ischemic stroke pathophysiology, including apoptosis, inflammation, oxidative stress, and neuronal damage, and their potential roles in diagnosis, treatment, and prognosis. The results section of this manuscript presents an in-depth review of the three principal miRNAs based on our lab research. Furthermore, we provide a summary of important findings from other stroke-related miRNAs in Appendix A.

### 4.2. Sampling and Enrollment

The study enrolled all patients who arrived at the adult Emergency Department within 6 h of witnessed stroke symptom onset. Inclusion required a confirmed diagnosis of acLVO by CT Angiogram. Healthy individuals formed the control group. Exclusion criteria included individuals with significant atherothrombotic disorders, such as pronounced coronary or peripheral vascular disease, deep vein thrombosis, or pulmonary embolism, patients with concurrent neurological conditions, a stroke in the preceding three months, posterior circulation LVO, uncertain time since last seen well, and pregnancy. Patients with recent thrombosis or severe atherosclerosis were excluded to avoid interference from clot-associated miRNAs, like platelet-derived miRNAs. Individuals with recent brain injuries were excluded to prevent confounding results from miRNA profiles linked to these recent injuries. Pregnant women were excluded due to potential variations in miRNA profiles in expectant mothers with growing fetuses.

### 4.3. Sample Collection and Processing

Blood samples were collected from acLVO patients at four time points: 0–6 h, 6–12 h, 12–24 h, and 5–7 days after symptom onset. Controls were healthy volunteers without any acute disease process or chronic stroke risk factors and had blood samples collected at a single time point. After initial sample collection and centrifugation, plasma was stored at −80 °C until further processing. Subsequently, EVs enriched in exosomes were isolated from the cell-free plasma using the Total Exosome Isolation Kit (Invitrogen, Thermofisher Scientific, Waltham, MA, USA) as previously described [50]. The isolated EVs were characterized for quantity and size using the NanoSight NS300 (Malvern Analytical, Malvern, UK) [51]. Total RNA was isolated using the Maxwell RSC miRNA tissue kit (Promega, Fitchburg, WI, USA) [52]. MiRNA profiling was performed using the multiplexed NanoString nCounter miRNA system (Nanostring Technologies, Seattle, WA, USA) as previously described [51]. Nanostring counts exported from nSolver were filtered and normalized with in-house scripts where miRNA and samples were filtered based on negative background probes (NegCutoffS1 = meanNegS1 + 1.5 × stdevNegS1) and normalized based on the geometric mean of expression and log2 transformed.

### 4.4. Statistical Analysis

Statistics were performed in R. Heatmaps with hierarchical clustering and principal component analysis (PCA) were performed with pheatmap and prcomp to visually group samples into clusters to give us an idea of the differences and similarities between samples and sample categories. Modest changes (<2 fold) in miRNA expression are known to be associated with changes in target gene expression [53,54,55]. Changes in miRNAs across time points in LVO patients were assessed using repeated measures ANOVA, and we tested time point differences between any two time points with a mixed model with a sample included as a random effect. The significance for any statistical test was defined as FDR < 0.05.

### 4.5. Ingenuity Pathway Analysis (IPA)

The IPA pathway analysis was performed on miRNAs with a statistical significance of *p* < 0.05 for each time-point comparison. The mRNA targets for these significantly altered miRNAs were identified, focusing on targets with experimental validation or high-confidence predictions. Using IPA Core Analysis, pathway enrichment was conducted to identify relevant canonical pathways. IPA calculated a *p*-value based on the overlap between the target gene list and the total genes annotated in each pathway. Pathways were compared between samples collected at 6 h post-stroke and those at later time points (12 h, 24 h, and 7 days) to observe temporal changes in pathway activation. For visualization, pathways associated with “Cancer” were excluded, and only pathways with –log10(p) values greater than 3.5 were displayed.

## 5. Conclusions

In conclusion, our investigation has shed light on the intricate role of miRNAs in stroke pathophysiology, highlighting their potential as biomarkers for acute cerebrovascular events. By identifying 11 miRNAs, particularly miR-140-5p, miR-7-5p, and miR-210-3p, with significant differential expression within 6 h of stroke onset, our study suggests these miRNAs could potentially serve as valuable indicators for diagnosis and possible targets for therapy, given their involvement in critical pathways like inflammation, oxidative stress, and angiogenesis. Despite promising indications for early detection and stroke management, the limitations of our study call for extensive validation through larger, risk-matched cohorts in multi-center trials. Such rigorous research is essential for confirming miRNAs’ utility as reliable clinical biomarkers and for potentially uncovering novel therapeutic strategies that could significantly improve patient outcomes. It is anticipated that the present findings will encourage further detailed exploration of miRNA functions post-ischemic stroke using animal models, fostering advancements in clinical approaches and patient care.

## Figures and Tables

**Figure 1 ijms-26-00747-f001:**
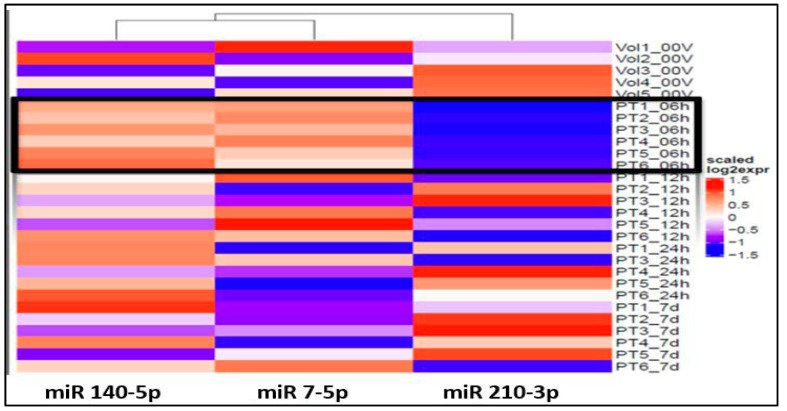
Heat map shows over-expression (red spectrum) of miRs 140-5p and 7-5p and under-expression of miR 210-3p (blue spectrum) within 6h of onset (PT#_06h): Note: 6h expression is marked by black outline.

**Figure 2 ijms-26-00747-f002:**
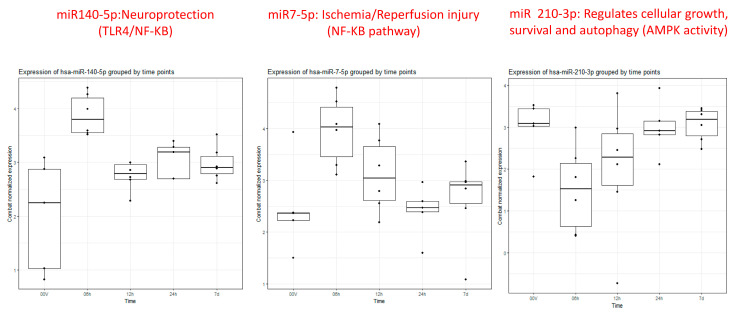
Box plots of miRs 140-5p, 7-5p, and 210-3p showing differential expression at 6 h (2nd box plot 0–6 h) compared to healthy controls (1st box plot) and remaining time points (3rd (6–12 h), 4th (12–24 h), and 5th (5–7 days) box plots).

**Figure 3 ijms-26-00747-f003:**
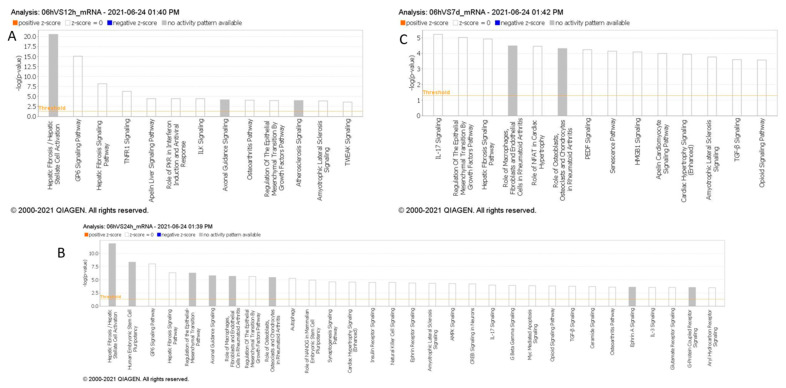
(**A**–**C**): Pathway analysis of differentially expressed miRNAs within 6 h compared to 6–12 h (**A**), 12–24 h (**B**), and 5–7 days (**C**) post-stroke. MiRNAs with *p* < 0.05 were analyzed to identify high-confidence mRNA targets, and canonical pathway enrichment was performed using IPA Core Analysis. Pathways with –log10(p) > 3.5 are shown, excluding cancer-related pathways.

**Figure 4 ijms-26-00747-f004:**
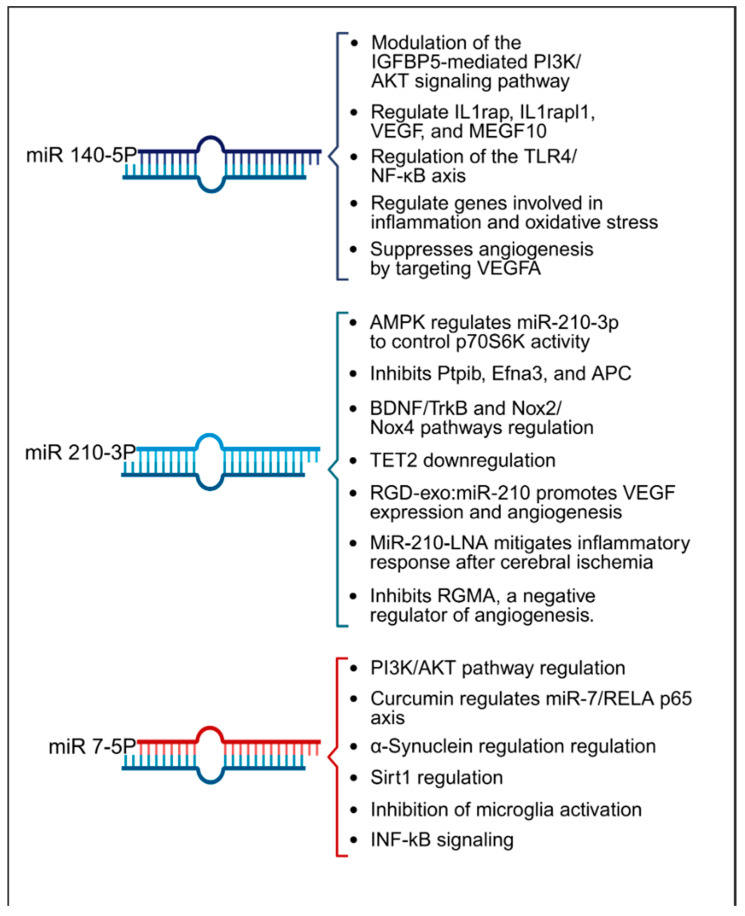
Summary of key pathways influenced by miRNAs 140-5p, 210-3p, and 7-5p, highlighting their roles in inflammation, angiogenesis, and oxidative stress within the context of stroke.

**Table 1 ijms-26-00747-t001:** Demographics and clinical characteristics of acLVO patients.

Variables/Patient ID	PT1	PT2	PT3	PT4	PT5	PT6
Age	85	57	45	56	43	14
Gender	Male	Female	Male	Male	Male	Male
Race	White	White	White	White	White	White
IV tPA (Yes/No)	Yes	Yes	Yes	Yes	Yes	Yes
Biosampling time (pre, intra, and post ET)	pre (TICI3)	pre (NT)	post (TICI2b)	pre (TICI2b)	pre (NT)	intra (TICI2b)
Etiology	Cryptogenic	Cryptogenic	Cryptogenic	Cardioembolic	LAA	LAA
HTN	Yes	Yes	No	Yes	No	No
HLD	Yes	Yes	No	Yes	No	No
DM	No	No	No	No	No	No
Atrial fib/Flutter	No	No	No	Yes	No	No
IV Drug use	No	No	No	No	No	No
Smoking	No	Yes	No	No	No	No
Hx of CAD	No	No	No	No	No	No
Anticoagulants	No	No	Yes	No	No	No
Antiplatelets	Yes	Yes	No	Yes	Yes	Yes
Statin Use	Yes	Yes	No	Yes	Yes	Yes
Site of Occlusion	R-ICA	R-MCA	L-MCA	R-MCA	L-ICA	L-MCA
NIHSS at presentation	17	3	17	10	4	1
NIHSS at discharge	8	10	10	10	8	3
MRS at discharge	4	3	4	2	1	2
HT	Yes	No	No	No	No	No

Hypertension (HTN), Hyperlipidemia (HLD), Diabetes Mellitus (DM), Atrial fibrillation/Atrial flutter (Atrial fib/flutter), NIH stroke scale (NIHSS), Modified Rankin Scale (MRS), Hemorrhagic Transformation (HT), Patient (PT).

## Data Availability

Data is contained within the article and Appendix A. Any additional information can be provided upon request by the corresponding author.

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
