# Peer review of "MicroRNA Expression Profile in Acute Ischemic Stroke"

_ijms, 2025, doi:10.3390/ijms26020747_

Round 1
Reviewer 1 Report
Comments and Suggestions for Authors
The manuscript presents a pilot study exploring the expression profiles of specific microRNAs (miRNAs) in acute ischemic stroke (AIS) with a focus on anterior circulation large vessel occlusion (acLVO). The study's design, objectives, and findings are relevant and contribute to the growing body of literature on biomarkers in stroke. However, several areas could be improved for clarity, depth, and impact.
The abstract provides a good overview.While the introduction outlines the clinical significance of acute ischemic stroke.
Figures 1-3 effectively visualize the data, but the legends could be expanded to explain the key takeaways in greater detail for readers who may not be familiar with miRNA studies.
Table 2, which summarizes existing literature, is thorough but would benefit from an additional column outlining the limitations or gaps in each study.
Ensure consistent use of technical terms, e.g., "miR" versus "microRNA." Maintaining uniformity avoids confusion and improves readability.
Include a graphical abstract or a summary figure that visually represents the main findings and pathways affected by miRNAs in AIS.
Reviewer 2 Report
Comments and Suggestions for Authors
Journal: IJMS (ISSN 1422-0067)
Manuscript ID: ijms-3382111
Type: Article
Title: MicroRNA Expression Profile in Acute Ischemic Stroke
Title title “MicroRNA Expression Profile in Acute Ischemic Stroke” by Shraddha Mainali et al, provide a promising advancement in our knowledge of the dynamic role of circulating EV-encapsulated miRNAs in acute large vessel occlusion (acLVO) ischemic stroke is offered by this study. MiR-140-5p, miR-7-5p, and miR-210-3p are among the 11 miRNAs that the study found to have notable temporal expression variations. This suggests that these miRNAs may be useful as early diagnostic indicators and treatment targets. In addition to highlighting the importance of important pathways like inflammation, oxidative stress, and angiogenesis in stroke pathophysiology, the data also show how useful longitudinal profiling is for capturing the changing molecular landscape after stroke onset.
I suggest few minor comments that authors need to address.
General suggestion:
1. Although the abstract discusses a study on extracellular vesicle (EV) miRNAs in acLVO stroke, it is unclear if the study's primary focus is on the miRNA expression profile at every stage of AIS. Although circulating miRNAs in a particular stroke subtype (acLVO) are the main focus of the abstract, the title implies a wide investigation of microRNA profiles in acute ischemic stroke. It would be helpful to provide a more comprehensive description of how these findings can be applied to other AIS subtypes in addition to acLVO in order to match the content with the headline.
Study design:
2. Could the authors explain the inclusion of both a cohort study and a scoping review? Would clarity be enhanced by splitting these elements?
3. Is there a special objective for focusing on miR-140-5p, miR-210-3p, and miR-7-5p? Were these miRNAs pre-selected based on past research or early findings?
4. Figure 1: Is the discussion of miRNA expression (overexpression of miR7-5p and under expression of miR210-3P) related to their observed biological significance in ischemia, reperfusion, or cellular processes? Could the author provide additional context from the literature?
5. Is there enough evidence to correlate miR-140-5P, mir-210-3P, and miR-7-5P to stroke-related processes (such as angiogenesis, oxidative stress, and microglia activation)?
Please mention in the text.
Discussion:
6. The authors report significant time-dependent changes in miR-140-5p, miR-7-5p, and miR-210-3p. Could the authors clarify how these temporal dynamics relate to clinical stroke severity (NIHSS score) or radiological findings (infarct volume or reperfusion status)?
Conclusion:
7. Highlighting the study's novelty reinforces its contribution to the field and strengthens the conclusion's impact.
Round 2
Reviewer 1 Report
Comments and Suggestions for Authors
The author has successfully addressed my comments, and I am pleased to accept this manuscript.